# Recognition Site Modifiable Macrocycle: Synthesis, Functional Group Variation and Structural Inspection

**DOI:** 10.3390/molecules28031338

**Published:** 2023-01-31

**Authors:** Linmeng Fan, Min Du, Lichun Kong, Yan Cai, Xiaobo Hu

**Affiliations:** Key Laboratory of the Ministry of Education for Advanced Catalysis Materials, College of Chemistry and Life Sciences, Zhejiang Normal University, 688 Yingbin Road, Jinhua 321004, China

**Keywords:** macrocyclic molecule, aromatic foldamer, post-synthesis modification, recognition site variation, structural inspection

## Abstract

Traditional macrocyclic molecules encode recognition sites in their structural backbones, which limits the variation of the recognition sites and thus, would restrict the adjustment of recognition properties. Here, we report a new oligoamide-based macrocycle capable of varying the recognition functional groups by post-synthesis modification on its structural backbone. Through six steps of common reactions, the parent macrocycle (**9**) can be produced in gram scale with an overall yield of 31%. The post-synthesis modification of **9** to vary the recognition sites are demonstrated by producing four different macrocycles (**10**–**13**) with distinct functional groups, 2-methoxyethoxyl (**10**), hydroxyl (**11**), carboxyl (**12**) and amide (**13**), respectively. The ^1^H NMR study suggests that the structure of these macrocycles is consistent with our design, i.e., forming hydrogen bonding network at both rims of the macrocyclic backbone. The ^1^H-^1^H NOESY NMR study indicates the recognition functional groups are located inside the cavity of macrocycles. At last, a preliminary molecular recognition study shows **10** can recognize n-octyl-β-D-glucopyranoside (**14**) in chloroform.

## 1. Introduction

In nature, the variety of the amino acid side chains is the key for protein to realize the recognition of small molecules and thereby access the corresponding bio-functions, such as carbohydrate protein [1,2,3] and sortase transpeptidases [4,5,6]. Inspired by nature, large amount of artificial molecules or complexes were synthesized to structurally or functionally mimic functional protein, for instance, foldamers [7,8,9,10,11,12,13,14,15], macrocycles [16,17,18,19,20,21,22], cages [23,24,25,26,27], hemicagepodates [28,29,30,31,32], etc. Wherein, macrocycles represent one of the most popular mimics. Thanks to their enhanced structural rigidity, macrocycles could exhibit improved metabolic stability and pharmaceutical activity comparing to acyclic biologics [33,34]. For example, biaryl-bridged macrocycles can provide high binding affinity and good target selectivity [35] and the access of these macrocycles can be realized by intramolecular Suzuki reaction [36,37], McMurry reaction [38], solid-supported synthesis [39] and oxidative coupling reactions [40]. In addition to pharmaceutical applications, there are even more applications in other bio-related fields, such as β-glucopyranose recognition [41], maltodextrin recognition [42], purine and pyrimidine bases recognition [43], oligopeptide recognition [44], water transmembrane transport [45], dye recognition for bio-imaging [46], drug delivery [47], etc. However, in most cases, the recognition sites of macrocycles, and most other host molecules, are unable to be varied because they are already encoded in the structural backbones of these molecules. This leads to a common defect of traditional macrocycles that their binding strength and selectivity are not adjustable. In addition, the limitation of varying recognition sites would also restrict the exploration of the recognition characteristics of many unexploited functional groups and the reasonable comparison of the binding properties of different functional groups in the same backbone. Herein, we propose a new macrocyclic backbone that is feasible to perform post-synthesis modification in its inner rim to achieve the variation of the recognition sites (Figure 1). The production of the parent macrocycle (**9**) can be realized through convenient reactions with an overall yield of 31%. The inner rim phenolic hydroxyl group of the parent macrocycle allows the installation of diverse functional groups inside the macrocycle in a high efficient way, which is evidenced by producing the exemplified macrocycles **11**–**13**, with 2-methoxyethoxyl (**10**), hydroxyl (**11**), carboxyl (**12**) and amide (**13**) functional groups, respectively. In the end, NMR studies gave insight into the structure of macrocycles and spatial location of the recognition functional groups. The exhibited ability of installing diverse functional groups inside a macrocycle could provide a rich imagination of the mimic of protein functions, such as molecular recognition and enzymatic catalysis.

## 2. Results and Discussion

The design of macrocyclic molecules often encodes recognition sites in the structural backbone, leading to the restriction of varying recognition functional groups. Incorporating recognition sites into side chains is a feasible strategy to address the above problem. For example, zinc porphyrin hosts can change their side chains by using different aromatic aldehydes or by post-synthesis modification, thus endowing these hosts with tunable binding affinity to nitrogen-containing bases [48,49]. We also used this strategy to access macrocycles with variable recognition sites. The backbone of the target macrocycles is designed as the combination of two tethers (for facilitating the post-synthesis modification of functional groups inside the cavity) and two spacers (for varying the size of the cavity). Through the inner rim hydrogen binding network, the tether (aromatic oligoamide section, Figure 1, in black) can be endowed with structural rigidity and an appropriate curvature to increase the possibility of the access of the desired macrocycle. Moreover, the tether can be produced through many combinations of small fractional units, such as the pyridine-benzene-pyridine (py-ben-py) combination shown here. Each fractional unit can be easily modified independently with the desired side chains on the outer rim for various purposes, such as for solubility in organic or aqueous solvent systems. More importantly, with the proposed aromatic oligoamide sequence, py-ben-py, various recognition functional groups can be easily installed on the inner rim of the parent macrocycle through simple substitution reactions (Figure 1 and Figure 2). On the other hand, the size of the cavity of the macrocycle is dependenton the spacer (here, a biphenyl unit, Figure 1, in plum). The spacer can provide π-π and dispersion interactions and hydrophobic interactions in the case of aqueous media. By varying the length of the spacer, the size of the cavity also could be adjusted for accommodating different sized guests. Furthermore, by increasing the connection sites of the spacer (by using 3,3′-diaminobenzidine, for example), it could easily achieve the transformation of macrocyclic molecules to cage molecules to access higher binding affinity and better selectivity to mimic the protein recognition pocket. 

The synthetic routes towards the parent macrocycle could be flexible and here shows one synthetic approach as an example (Figure 1). First, the precursor of the middle unit of the tether, 2-methoxy-2-oxoethyl 3,5-diamino-4-hydroxybenzoate (**3**), was prepared from the commercial compound **1**. By applying a mild reaction condition and limit amount of methyl 2-bromoacetate, selective substitution on the carboxyl group could be achieved with a high yield of 90%. Afterwards, the nitro group of the resulting compound **2** was quantitatively converted to amino groups by hydrogenation under the catalysis of 10% Pd/C to give the target precursor, compound **3**. It should be noted that diamino-4-hydroxybenzoate is ready to be oxidized and we had experienced many failures before the present version with methyl ester side chain. The pyridine unit of the tether was prepared as the mono-ester form, compound **5**, through saponification reaction. Then, the carboxyl acid group of compound **5** was converted to acid chloride by oxalyl chloride followed by the addition of the spacer precursor **6** to afford the diester compound **7**. Afterwards, saponification of compound **7** was performed to disclose the carboxyl acid group, compound **8**, and it was ready to conduct amino acid coupling (cyclization) with the previously prepared diamino compound **3**. The final step [2+2] cyclization under the coupling reagent PyBop was very efficient (57% yield) thanks to the encoded hydrogen bonding network which can guide the fragments to be assembled in a desired way. Thereby, the present synthetic approach allows the production of the parent macrocycle **9** in a very efficient way, only one column chromatography, in less than 10 days and can be produced in gram scale with the overall yield of 31%.

The variation of the recognition functional groups inside the parent macrocycle **9** can be easily achieved through substitution reactions. Here, we show the installation of four different functional groups as an example (Figure 2), 2-methoxyethoxyl group (**10**) to represent the ethylene glycol functional groups, and hydroxyl (**11**), carboxyl (**12**) and amide (**13**) groups to mimic serine, aspartic acid and asparagines side chains, respectively. The installation of 2-methoxyethoxyl and hydroxyl groups is very straightforward. Through one step substitution reaction on the phenolic hydroxyl groups of **9** with the corresponding bromide, the target macrocycles **10** and **11** can be produced in good yields, 47% and 41%, respectively. The production of macrocycle **12** with carboxyl groups required two steps, a similar substitution reaction with tert-butyl bromoacetate and followed by the deprotection of the tert-butyl group with trifluoroacetic acid. Although it required one step more, the two-steps overall yield was even higher, reaching a yield of 63%. Based on macrocycle **12**, the alternation to the amide group can be easily achieved. Through acylation reaction with oxalyl chloride, the carboxyl group was converted to the acid chloride. Afterwards, by the addition of ammonium, the macrocycle with the amide groups (**13**) can be produced with an overall yield of 28%, starting from the parent macrocycle **9**. Additionally, we had also attempted to install the amide group directly onto the parent macrocycle by using the corresponding bromide. However, the yield was much lower and the separation was difficult.

Afterwards, ^1^H NMR studies were conducted to evidence the formation of the designed hydrogen bonding network. Here, we take macrocycle **10** as an example. ^1^H NMR spectra of compounds **2** and **10** in CDCl_3_ show the benzene protons (**H_ben_**, Figure 3, in green) are at 9.00 and 9.17 ppm, respectively. It is reasonable for the **H_ben_** of **2** to have significant downfield shift since there are two nitro groups nearby. However, the **H_ben_** of **10** are even more downfield shifted. This is probably due to the formation of a rather rigid structure of compound **10** favoring the **H_ben_** of **10** to form rather stronger hydrogen bonding with the nearby oxygen atoms. Similar observations can also be found on the pyridine protons (**H_py_**, Figure 3, in blue). These observations in turn imply the hydrogen bonding network is formed in the designed manner. Moreover, the presence of two distinct amide signals (**H_NH_**), one at 10.34 ppm and the other at 9.35 ppm in the ^1^H NMR spectrum of **10** (Figure 3a, in red) is also in good agreement of the designed hydrogen bonding network that one **H_NH_** can form two hydrogen bonds with the nearby nitrogen and oxygen atoms while the other **H_NH_** can only form one hydrogen bond with the nearby nitrogen atom. Thereby, we proved the backbone structure of the macrocycle is formed as we proposed and, in turn, cluing the orientation of the installed functional groups should be towards the cavity of the macrocycle.

In order to gain further information of the structure, especially the location of the installed functional groups, 2D NMR studies were performed. Here, we also take macrocycle **10** as an example. First, ^1^H-^1^H NOESY spectrum of **10** in CDCl_3_ clearly shows the spatial correlation between **H_NH1_** and the methyl protons of the spacer (**H_NH1_**↔**H_M_**, Appendix A), yet the correlation **H_NH2_**↔**H_M_** is not observed, which evidences **H_NH2_** is the **H_NH_** more remote to **H_M_**. This is consistent with the above conclusion drawn from ^1^H NMR chemical shift. Similarly, other relevant correlations, such as **H_NH2_**↔**H_ben_** (Appendix A), can also be observed. Then, the protons of the installed 2-(2-methoxyethoxy)ethyl group can be assigned with the help of ^1^H-^1^H NOESY and ^1^H-^1^H COSY spectra (Appendix A). For instance, correlations **H_NH2_**↔**H_5_**, **H_5_**↔**H_4_**, **H_4_**↔**H_3_** and **H_3_**↔**H_2_** can be observed sequentially indicating **H_5_** is the one closest to the backbone while **H_2_** is remote to the backbone. With the identification of these key protons, the spatial correlations between the protons of the installed functional group and the protons of the parent macrocycle can be rationalized. For example, the presence of correlations **H_M_**↔**H_5_**, **H_M_**↔**H_4_** and **H_M_**↔**H_3,_** in addition with the absence of correlations **H_M_**↔**H_2_** and **H_M_**↔**H_1_** (Figure 4), indicate that the 2-methoxyethoxyl group has part of the structure located in the cavity and the methoxy terminal part is located outside of the cavity. This observation is in good agreement with the size of the 2-methoxyethoxyl group. It would cause severe steric hindrance between two 2-methoxyethoxyl groups if whole chains were located inside the cavity. Similarly, other relevant correlations, such as **H_NH2_**↔**H_5_**, **H_NH2_**↔**H_4_**, **H_NH1_**↔**H_5_**, **H_NH1_**↔**H_4_** and **H_NH1_**↔**H_3,_** can be observed as well (Figure 4). Since the length of the functional groups of macrocycles **11**–**13** is much shorter (i.e., hydroxyl, carboxyl and amide groups, respectively), it is reasonable to deduce that these functional groups should also be located inside the cavity of the parent macrocycle. Therefore, all NMR results evidence the formation of the designed structure.

At last, a preliminary molecular recognition study was conducted. By adding n-octyl-β-D-glucopyranoside (**14**) to the CDCl_3_ solution of **10**, the chemical shifts of **H_NH1_** and biphenyl (**H_BP_**) protons of **10** were observed (Figure 5, in red and blue), indicating the macrocycle **10** can recognize compound **14**. In addition, the chemical shifts of some protons at the outer rim of the backbone were also observed, such as **H_ben_** (Figure 5, in green), implying the macrocycle should twist its backbone to adapt the shape of the guest. However, due to the steric hindrance between the 2-methoxyethoxyl side chains of **10** and the octyl side chain of **14**, the binding strength between **10** and **14** is weak. In the future, we will screen diverse guest molecules to achieve higher binding affinity and selective recognition with the macrocycle **14**.

## 3. Materials and Methods

### 3.1. General Information

All commercially available starting materials and reagents were used without further purification. Anhydrous THF and DCM were obtained from commercial sources. Analytical thin layer chromatography (TLC) was performed on silica gel plates (Merck 60F254) visualized with a UV lamp (254 nm). Column chromatography was performed with commercial glass columns using silica gel 200–300 mesh (particle size 0.045–0.075 mm). High resolution electrospray ionization time-of-flight (HRESI-TOF) mass spectra were measured in the positive ion mode on an Agilent 6230 mass spectrometer.

The ^1^H NMR spectra were recorded on a Bruker Avance III HD 400 in CDCl_3_ or DMSO-*d*_6_. Chemical shifts are reported in ppm relative to the residual solvent signal of CDCl_3_ (*δ* = 7.26 ppm) or DMSO-*d*_6_ (*δ* = 2.50 ppm). Abbreviations used for signal multiplicity are: s = singlet, d = doublet, t = triplet, q = quartet, m = multiplet or overlap of nonequivalent resonances, br = broad. Coupling constants, *J*, are reported in Hertz (Hz). ^13^C{^1^H} NMR spectra were recorded on a Bruker AVANCE III HD 400 in CDCl_3_ or DMSO-*d*_6_ and the observed signals are reported in ppm relative to the residual solvent signal of CDCl_3_ (*δ* = 77.16 ppm) or DMSO-*d*_6_ (*δ* = 39.52 ppm).

Spectra of nuclear overhauser effect spectroscopy (NOESY) and correlation spectroscopy (COSY) experiments were recorded on a Bruker Avance III HD 400 by means of a BBO (BB-H/F-D) probe. Data processing was performed with Topspin software. ^1^H-^1^H NOESY acquisition of **10** was performed with a time domain size of 2048 (F2) × 256 (F1), 16 scans per increment, spectral width of 11.79 (F2) × 11.79 (F1) ppm, offset of 5.61 (F2) × 5.61 (F1), a pulse program of noesygpphpp, dwell time of 106 μs, relaxation delay of 2.04 s and a mixing time of 300 ms. ^1^H-^1^H COSY acquisition of **10** was performed with a time domain size of 2048 (F2) × 128 (F1), 4 scans per increment, spectral width of 11.79 (F2) × 11.79 (F1) ppm, offset of 5.61 (F2) × 5.61 (F1), dwell time of 106 μs, relaxation delay of 1.98 sand a pulse program of cosygpmfqf.

### 3.2. Preparation of 2-Methoxy-2-oxoethyl 4-hydroxy-3,5-dinitrobenzoate *(**2**)*

4-hydroxy-3, 5-dinitrobenzoic acid (compound **1**, 11.4 g, 50 mmol) and K_2_CO_3_ (10.36 g, 75 mmol) were suspended in 200 mL DMF, followed by the addition of methyl 2-bromoacetate (7.1 mL, 75 mmol) at 50 °C. The mixture was stirred at this temperature for 12 h and cooled to room temperature. The reaction mixture was concentrated under vacuum and the residue was treated with water to allow the precipitation of the crude. The solid was filtered, dried and washed with methanol to afford the titled compound as a yellow solid (13.5 g, 90%); ^1^H NMR (400 MHz, CDCl_3_) *δ* (ppm) = 11.82 (br, 1H), 9.00 (s, 2H), 4.93 (s, 2H), 3.82 (s, 3H); ^13^C{^1^H} NMR (101 MHz, CDCl_3_): *δ* (ppm) = 167.4, 162.0, 152.5, 137.5, 132.4, 120.7, 61.8, 52.6. HRMS (ESI): m/z calcd for C_10_H_8_N_2_O_9_Na [M + Na]^+^: 323.0122, found: 323.0118.

### 3.3. Preparation of 2-Methoxy-2-oxoethyl 3,5-diamino-4-hydroxybenzoate *(**3**)*

Compound **2** (5.1 g, 17 mmol) and 10% Pd/C (0.51 g) were suspended in 100 mL anhydrous THF under N_2_. Then, N_2_ was exchanged with H_2_ and the mixture was allowed to stir at room temperature for 24 h. Afterwards, Pd/C was filtered and the solution was concentrated under vacuum to afford the titled compound as a dark green sticky solid (4.1 g, quantitative). Note: compound **3** was ready to be oxidized and, thus, it was immediately used in the next step without further purification and characterization. 

### 3.4. Preparation of the Diester *(**7**)*

Dimethyl 4-(octyloxy)pyridine-2,6-dicarboxylate (compound **4**, 8.1 g, 25 mmol) was dissolved in 50 mL THF followed by the addition of 5 M NaOH aqueous solution (5 mL, 25 mmol). The mixture was stirred at room temperature for 12 h. THF was removed under vacuum and the reaction mixture was acidified by 1 M HCl. The resulting precipitation was collected and the solid was dried to give the crude of the intermediate **5**. Then, the intermediate **5** was dissolved in 100 mL anhydrous DCM followed by the addition of oxalyl chloride (4.23 mL, 50 mmol) under N_2_. The solution was stirred at room temperature for four hours. The solvent and extra oxalyl chloride were removed under vacuum and re-dissolved in 50 mL anhydrous THF. Compound **6** (2.4 g, 10 mmol) and DIPEA (8.3 mL, 50 mmol) were dissolved in another 50 mL anhydrous THF. The two THF solutions were then mixed immediately and the mixture was stirred at room temperature under N_2_ for 1 h. Solvent was removed under vacuum and the crude product was washed with methanol and ethyl acetate to give the titled compound as a white solid (5.6 g, 68%). ^1^H NMR (400 MHz, CDCl_3_) *δ* (ppm) = 9.66 (s, 2H), 7.97 (d, *J* = 2.0 Hz, 2H), 7.78 (d, *J* = 2.0 Hz, 2H), 7.35 (s, 4H), 4.17 (t, *J* = 6.4 Hz, 4H), 4.01 (s, 6H), 2.36 (s, 12H), 1.85 (m, 4H), 1.48 (m, 4H), 1.30 (m, 16H), 0.89 (t, *J* = 6.4 Hz, 6H); ^13^C{^1^H} NMR (101 MHz, CDCl_3_): *δ* (ppm) = 167.7, 165.1, 161.9, 151.9, 148.1, 139.8, 135.6, 132.9, 127.0, 115.0, 110.9, 69.1, 52.9, 31.8, 29.22, 29.19, 28.8, 25.9, 22.7, 18.8, 14.1; HRMS (ESI): *m*/*z* calcd for C_48_H_63_N_4_O_8_ [M + H]^+^: 823.4640 found: 823.4643.

### 3.5. Preparation of the Diacid *(**8**)*

Compound **7** (4.1 g, 5 mmol) was dissolved in 20 mL THF and followed by the addition of 5 M LiOH aqueous solution (3 mL, 15 mmol). The mixture was stirred at room temperature for one hour and THF was removed under vacuum. The residue was acidified by 1 M HCl and the resulting solid was filtered and dried to give the titled compound as a white solid (3.6 g, 90%); ^1^H NMR (400 MHz, DMSO-*d*_6_) *δ* (ppm) = 12.98 (br, 2H), 10.72 (s, 2H), 7.82 (d, *J* = 2.4 Hz, 2H), 7.76 (d, *J* = 2.4 Hz, 2H), 7.53 (s, 4H), 4.27 (t, *J* = 6.4 Hz, 4H), 2.28 (s, 12H), 1.78 (m, 4H), 1.44 (m, 4H), 1.27 (m, 19H, overlapped with grease signal), 0.86 (t, *J* = 6.8 Hz, 6H). ^13^C{^1^H} NMR (101 MHz, 10% CD_3_OD/CDCl_3_): *δ* (ppm) = 167.9, 166.5, 162.7, 151.6, 148.3, 139.7, 135.9, 133.0, 126.8, 114.4, 111.8, 69.2, 31.7, 29.2, 28.7, 25.8, 22.6, 18.4, 14.0. HRMS (ESI): *m*/*z* calcd for C_46_H_59_N_4_O_8_ [M + H]^+^: 795.4327 found: 795.4323.

### 3.6. Synthesis of the Macrocycle *(**9**)*

Compound **3** (0.6 g, 2.5 mmol), compound **8** (2.0 g, 2.5 mmol), PyBop (3.9 g, 7.5 mmol), and DIPEA (1.7 mL, 10 mmol) were dissolved in 200 mL DMF. The mixture was stirred at 80 °C for 12 h. The reaction mixture was concentrated under vacuum and followed by the addition of 1M HCl to adjust the pH to 1–2. The resulting solid was collected by filtration and purified by column chromatography (eluent: DCM/MeOH = 10:1 *v*/*v*). The crude product was further washed with methanol to afford the titled compound as a yellow solid (1.43 g, 57%); ^1^H NMR (400 MHz, DMSO-*d*_6_) *δ* (ppm) = 10.88 (s, 4H), 10.20 (s, 4H), 8.79 (s, 4H), 7.85 (s, 4H), 7.74 (s, 4H), 7.38 (s, 8H), 4.81 (s, 4H), 4.28 (s, 8H), 3.71 (s, 6H), 2.26 (s, 24H), 1.79 (s, 8H), 1.46 (m, 58H, overlapped with grease signal), 0.87 (m, 14H, overlapped with grease signal); ^13^C{^1^H} NMR (101 MHz, 10% DMSO-*d*_6_/CDCl_3_): *δ* (ppm) = 169.4, 167.8, 167.1, 162.6, 161.2, 151.7, 151.2, 139.3, 136.2, 133.8, 127.2, 126.6, 118.0, 116.1, 111.3, 111.2, 68.9, 60.6, 52.1, 40.2, 39.8, 39.4, 36.4, 31.6, 29.5, 29.1, 28.7, 25.7, 22.5, 18.5, 14.0. HRMS (ESI): *m*/*z* calcd for C_112_H_134_N_12_O_22_ [M + 2H]^2+^: 999.9885 found: 999.9885.

### 3.7. Synthesis of the Macrocycle *(**10**)*

Compound **9** (100 mg, 0.05 mmol), K_2_CO_3_ (21 mg, 0.15 mmol) were suspended in 5 mL DMF, followed by the addition of 1-bromo-2-(2-methoxyethoxy)ethane (0.34 mL, 0.25 mmol). The mixture was stirred at 80 °C for 12 h. Then, 1 M HCl aqueous solution was added and the resulting solid was collected by filtration. The crude product was further purified by column chromatography (eluent: DCM/MeOH = 50:1 *v*/*v*) to afford the titled compound as a pale pink solid (52 mg, 47%); ^1^H NMR (400 MHz, CDCl_3_) *δ* (ppm) = 10.34 (s, 4H), 9.35 (s, 4H), 9.17 (s, 4H), 8.04 (d, *J* = 2.4 Hz, 4H), 7.99 (d, *J* = 2.4 Hz, 4H), 7.39 (s, 8H), 4.92 (s, 4H), 4.36 (br, 4H), 4.21 (t, *J* = 6.4 Hz, 8H), 3.82 (s, 6H), 3.60 (br, 4H), 3.38–3.26 (m, 4H), 3.15–3.06 (m, 4H), 2.99 (s, 6H), 2.31 (s, 24H), 1.87 (m, 8H), 1.52–1.21 (m, 49H, overlapped with water signal), 0.89 (t, *J* = 6.4 Hz, 12H); ^13^C{^1^H} NMR (101 MHz, CDCl_3_): *δ* (ppm) = 168.4, 168.2, 165.2, 161.8, 161.4, 150.9, 150.6, 140.9, 140.0, 135.6, 132.9, 131.6, 127.2, 126.5, 118.6, 112.2,112.1, 71.6, 70.0, 69.4, 68.7, 61.3, 58.6, 52.3, 31.8, 29.4, 29.2, 29.16, 28.8, 25.8, 22.7, 18.6, 14.1. HRMS (ESI): *m*/*z* calcd for C_122_H_154_N_12_O_26_ [M + 2H]^2+^: 1102.0566 found:1102.0539.

### 3.8. Synthesis of the Macrocycle *(**11**)*

Compound **9** (100 mg, 0.05 mmol) and K_2_CO_3_ (21 mg, 0.15 mmol) were suspended in 5 mL DMF, followed by the addition of 2-bromoethanol (0.18 mL, 0.25 mmol). The mixture was stirred at 80 °C for 12 h. Then, 1 M HCl aqueous solution was added and the resulting solid was collected by filtration. The crude product was further purified by column chromatography (eluent: DCM/MeOH = 50:1 *v*/*v*) to afford the titled compound as a white solid (43 mg, 41%); ^1^H NMR (400 MHz, DMSO-*d*_6_) *δ* (ppm) = 11.19 (s, 4H), 10.76 (s, 4H), 8.85 (s, 4H), 7.89 (d, *J* = 2.4 Hz, 4H), 7.78 (m, 12H), 6.02 (t, *J* = 4.8 Hz, 2H), 5.01 (s, 4H), 4.28 (t, *J* = 6.0 Hz, 8H), 4.16 (br, 4H), 3.74 (s, 6H), 2.24 (s, 24H), 1.78 (m, 8H), 1.43 (m, 8H), 1.26 (m, 34H, overlapped with grease signal), 0.85 (t, *J* = 6.4 Hz, 13H, overlapped with grease signal); ^13^C{^1^H} NMR (101 MHz, 2%DMSO-*d*_6_/CDCl_3_): *δ* (ppm) = 168.4, 168.1, 165.1, 162.4, 161.8, 151.3, 150.4, 142.7, 139.7, 136.1, 133.5, 131.4, 126.9, 125.7, 118.9, 112.1, 111.8, 69.3, 61.3, 60.6, 52.2, 31.8, 29.2, 29.2, 28.8, 25.8, 22.6, 18.6, 14.1. HRMS (ESI): m/z calcd for C_116_H_142_N_12_O_24_ [M + 2H]^2+^: 1044.0147 found: 1044.0139.

### 3.9. Synthesis of the Macrocycle *(**12**)*

Compound **9** (200 mg, 0.1 mmol), K_2_CO_3_ (42 mg, 0.3 mmol) were suspended in 10 mL DMF, followed by the addition of tert-butyl bromoacetate (0.73 mL, 0.50 mmol). The mixture was stirred at 80 °C for 12 h. Then, 1 M HCl aqueous solution was added and the resulting solid was collected by filtration and was further purified by column chromatography (eluent: DCM/MeOH = 50:1 *v*/*v*). The intermediate was then treated with TFA (1 mL) in DCM (4 mL) for 12 h. The reaction mixture was concentrated under vacuum and then washed by ether and methanol to afford the titled compound as a white solid (133 mg, 63%); ^1^H NMR (400 MHz, DMSO-*d*_6_) *δ* (ppm) = 12.41 (s, 4H), 11.40 (s, 4H), 8.82 (s, 4H), 7.85 (s, 4H), 7.77 (s, 4H), 7.64 (s, 8H), 5.00 (s, 4H), 4.28 (br, 8H), 4.20 (br, 4H), 3.74 (s, 6H), 2.24 (s, 24H), 1.79 (br, 8H), 1.45 (br, 8H), 1.27 (m, 45H, overlapped with grease signal), 0.86 (m, 14H, overlapped with grease signal). ^13^C{^1^H} NMR (101 MHz, 2%DMSO-*d*_6_/CDCl_3_): *δ* (ppm) = 168.3, 168.2, 165.1, 162.5, 162.0, 151.1, 150.7, 139.3, 139.1, 136.4, 133.6, 130.8, 126.6, 126.5, 112.0, 112.3, 112.29, 111.9, 69.2, 61.3, 52.3, 31.8, 29.7, 29.2, 29.19, 28.8, 25.8, 22.6, 18.6, 14.1. HRMS (ESI): *m*/*z* calcd for C_116_H_138_N_12_O_26_ [M + 2H]^2+^: 1057.9940 found:1057.9937. 

### 3.10. Synthesis of the Macrocycle *(**13**)*

Compound **12** (73 mg, 0.034 mmol) was suspended in 5 mL DCM. Oxalyl chloride (0.15 mL, 0.173 mmol) was added at room temperature. The mixture was stirred at this temperature for 4 h. DCM and the extra oxalyl chloride were removed under vacuum and re-dissolved with 5 mL DCM. Ammonium hydroxide (0.06 mL, 0.173 mmol) was then added and the mixture was stirred at room temperature for one hour. Solvents were removed under vacuum and the resulting solid was washed with methanol to give the titled compound as a pale yellow solid (32 mg, 44%); ^1^H NMR (400 MHz, DMSO-*d*_6_) *δ* (ppm) = 11.56 (s, 4H), 10.79 (s, 4H), 8.88 (s, 4H), 7.91 (s, 4H), 7.81 (s, 4H), 7.73 (s, 8H), 5.01 (br, 4H), 4.63 (br, 4H), 4.29 (br, 8H), 3.74 (s, 6H), 2.24 (s, 24H), 1.79 (br, 8H), 1.44 (br, 8H), 1.27 (m, 42H, overlapped with grease signal), 0.86 (m, 17H, overlapped with grease signal). ^13^C{^1^H} NMR (101 MHz, 5%DMSO-*d*_6_/CDCl_3_): *δ* (ppm) = 170.8, 168.3, 168.2, 165.3, 162.1, 151.2, 150.5, 139.6, 136.1, 133.4, 131.1, 126.6, 126.0, 119.4, 112.1, 112.0, 69.2, 61.3, 52.3, 31.7, 29.6, 29.2, 29.19, 29.14, 28.7, 25.8, 22.6, 18.4, 14.0. HRMS (ESI): *m*/*z* calcd for C_116_H_139_N_14_O_24_Na [M + H + Na]^2+^: 1068.0009 found:1067.9991.

### 3.11. ^1^H NMR, ^13^C{^1^H} NMR, Mass, ^1^H-^1^H COSY and ^1^H-^1^H NOESY NMR Spectra

^1^H NMR, ^13^C{^1^H} NMR, mass, ^1^H-^1^H COSY and ^1^H-^1^H NOESY NMR spectra are documented in Appendix A.

## 4. Conclusions

In conclusion, we have developed a synthetic strategy to produce a new macrocycle with modifiable recognition sites. The present synthetic approach allows producing the parent macrocycle and various macrocycles with different functional groups in an efficient way. The ^1^H NMR and ^1^H-^1^H NOESY NMR studies evidence the formation of the designed hydrogen bonding network and prove the location of the installed functional groups inside the cavity of the macrocycle. We envision the proposed molecular structure and synthetic protocol would inspire other designs to access the host molecules with variable recognition sites. In the future, we will optimize the size of the cavity and screen more functional groups for the purpose of specific recognition of bio-active targets.

## Data Availability

Data are contained within the article and Appendix A.

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
