# Peer review of "Recognition Site Modifiable Macrocycle: Synthesis, Functional Group Variation and Structural Inspection"

_molecules, 2023, doi:10.3390/molecules28031338_

Round 1

Reviewer 1 Report

In the manuscript entitled “Recognition site modifiable macrocycle: synthesis, functional group variation and structural inspection” by Hu and co-workers, they employed a convergent route to prepare a macrocyclic compound, which then further converted it to several derivatives. Then, the authors investigated the hydrogen-bonding network through 1H and NOESY NMR studies. The synthetic route toward the parent macrocycle is concise, and the conclusion based on the NMR study is correct. However, the authors have not shown any recognition study on those macrocyclic molecules in this manuscript, making this work incomplete and, particularly, inconsistent with the article title. To this one, this is a nice work and should be acceptable by Molecules. The only suggestion is to provide couple of examples on recognition study of certain molecules. 

Author Response

Point 1:  In the manuscript entitled “Recognition site modifiable macrocycle: synthesis, functional group variation and structural inspection” by Hu and co-workers, they employed a convergent route to prepare a macrocyclic compound, which then further converted it to several derivatives. Then, the authors investigated the hydrogen-bonding network through 1H and NOESY NMR studies. The synthetic route toward the parent macrocycle is concise, and the conclusion based on the NMR study is correct.

Response 1:  Thank you very much for thoroughly examining our manuscript.

Point 2:  However, the authors have not shown any recognition study on those macrocyclic molecules in this manuscript, making this work incomplete and, particularly, inconsistent with the article title. To this one, this is a nice work and should be acceptable by Molecules. The only suggestion is to provide couple of examples on recognition study of certain molecules. 

Response 2:  Thank you very much for the support and suggestions. Accordingly, we have added a preliminary molecular recognition study at the last part of the manuscript, which shows macrocycle 10 can recognize n-octyl-β-D-glucopyranoside (14) in chloroform. In the future, we will systematically screen diverse guest molecules as well as functional groups to achieve higher binding affinity and selective recognition, and the relevant results will be reported at an appropriate time.

Reviewer 2 Report

In this manuscript titled "Recognition site modifiable macrocycle: synthesis, functional group variation and structural inspection " - the methods of  1D and 2D NMR spectra of Recognition site modifiable macrocycle is used. The contents of the reviewed manuscript are very well described by the title. The authors show that the present synthetic approach allows producing the parent macrocycle as well as various macrocycles with different functional groups in an efficient way.
In my point, the main message of this manuscript is very nice sound and could be recommended for publication in Molecules after a corresponding revision in which the following issues are addressed:
-The authors have to shed light on the similarities and differences among their work and the literatures of the problem.
- You should discuss earlier work related to molecular recognition by macrocycles For exemple: 10.1080/10610278.2016.1238473 and 10.1021/ja00043a016 The designations shown in Figure 4 are not clear. Please separate the cross peaks of the NOESY and COSY spectra  and the  corresponding correlations in the structure by colors. For COSY cross peaks indicate couplings between two multiplets up to three, or occasionally four, bonds away. But for your picture view it looks like more than four bonds.
-A methods section should be extended to include more details about NMR experiments - the parameters of pulse sequence (width of spectra, values of relaxation delays, etc ) used must be described.
The scientific writing of this manuscript must be improved.
Page 2, Line 47. gave the insight of the --> gave insight into the
Page 2, Line 49. inside macrocycle --> inside a macrocycle
Page 2, Line 49. provide rich --> provide a rich
Page 2, Line 52. macrocyles --> macrocycles
Page 2, Line 65. depended --> dependent
Page 3, Lines 83-84. many times of failures --> many failures
Page 3, Line 88. saponificated --> saponification
Page 4, Line 114. attempt --> attempted
Page 4, Line 123-124. compound 2 have significant --> 2 to have significant
Page 4, Line 126. of rather --> of a rather
Page 4, Line 121. macrocycle 10 as the example --> macrocycle 10 as an example
Page 5, Line 159. if whole chains located inside ---> chains were located inside
Page 5, Lines 163-164. should also locate inside --> should also be located inside the
Page 6, Line 173. were obtained from commercial source --> were obtained from commercial sources
Page 9, Lines 325-326.functional groups are inside the cavity --> functional groups inside the cavity
It is the authors’ responsibility to improve the scientific writing through the whole manuscript.
If the authors submit a new version of manuscript according to my suggestions with more explanations about the experimental details and additional discussions, I could recommend the manuscript for publication.

Author Response

Point 1:  In this manuscript titled "Recognition site modifiable macrocycle: synthesis, functional group variation and structural inspection " - the methods of  1D and 2D NMR spectra of Recognition site modifiable macrocycle is used. The contents of the reviewed manuscript are very well described by the title. The authors show that the present synthetic approach allows producing the parent macrocycle as well as various macrocycles with different functional groups in an efficient way.
In my point, the main message of this manuscript is very nice sound and could be recommended for publication in Molecules after a corresponding revision in which the following issues are addressed:

Response 1:  Thank you very much for the support and your time spent on thoroughly examining our manuscript.

Point  2:  The authors have to shed light on the similarities and differences among their work and the literatures of the problem.

Response 2: We added examples of other biaryl-bridged macrocycles and their synthetic strategies in the Introduction part. Regarding the problem of the reported macrocycles, we have also stated in the Introduction section.

Point  3:  You should discuss earlier work related to molecular recognition by macrocycles For exemple: 10.1080/10610278.2016.1238473 and 10.1021/ja00043a016

Response 3: We added the suggested examples at the beginning of the Results and Discussion section to show that incorporating recognition sites into side chains is a feasible strategy to vary the recognition functional groups on hosts and thus enables the modulation of recognition properties of zinc porphyrin hosts.

Point  4:  The designations shown in Figure 4 are not clear. Please separate the cross peaks of the NOESY and COSY spectra  and  the  corresponding correlations in the structure by colors.

Response 4: We separated NOESY and COSY spectra and they are now shown individually in Figure 4, and Figures S1-5. The corresponding correlations in Figure 4 are now shown by different colors.

Point  5:   For COSY cross peaks indicate couplings between two multiplets up to three, or occasionally four, bonds away. But for your picture view it looks like more than four bonds.

Response 5: Regarding the doubt of the cross peaks in COSY, we honestly do not find couplings through more than four bonds. Maybe the previously stacked NOESY spectrum distracts attentions. Now, we have separated these spectra and COSY is shown in Figures S2 and S5.

Point  6:  A methods section should be extended to include more details about NMR experiments - the parameters of pulse sequence (width of spectra, values of relaxation delays, etc ) used must be described.

Response 6: we added more experimental details in the Materials and Methods section.

Point  7:  The scientific writing of this manuscript must be improved.
Page 2, Line 47. gave the insight of the --> gave insight into the
Page 2, Line 49. inside macrocycle --> inside a macrocycle
Page 2, Line 49. provide rich --> provide a rich
Page 2, Line 52. macrocyles --> macrocycles
Page 2, Line 65. depended --> dependent
Page 3, Lines 83-84. many times of failures --> many failures
Page 3, Line 88. saponificated --> saponification
Page 4, Line 114. attempt --> attempted
Page 4, Line 123-124. compound 2 have significant --> 2 to have significant
Page 4, Line 126. of rather --> of a rather
Page 4, Line 121. macrocycle 10 as the example --> macrocycle 10 as an example
Page 5, Line 159. if whole chains located inside ---> chains were located inside
Page 5, Lines 163-164. should also locate inside --> should also be located inside the
Page 6, Line 173. were obtained from commercial source --> were obtained from commercial sources
Page 9, Lines 325-326.functional groups are inside the cavity --> functional groups inside the cavity
It is the authors’ responsibility to improve the scientific writing through the whole manuscript.

Response 7: Thank you again for your careful reading. We corrected the writing mistakes according to the suggestions and tried our best to correct more.

Point  8:  If the authors submit a new version of manuscript according to my suggestions with more explanations about the experimental details and additional discussions, I could recommend the manuscript for publication.

Response 8: Thank you again for your support and suggestions. As you can see from the above responses, we revised the manuscript according to your suggestions and added more information. In addition, we added a preliminary molecular recognition study at the last part of the manuscript, which shows macrocycle 10 can recognize n-octyl-β-D-glucopyranoside (14) in chloroform. In the future, we will systematically screen diverse guest molecules as well as functional groups to achieve higher binding affinity and selective recognition, and the relevant results will be reported at an appropriate time.

Reviewer 3 Report

The paper is devoted to the synthesis of new macrocyclic compound with strong H-
bonding like protein-mimic. The manuscript describes synthesis and
characterization of four new macrocycles. In that part I do not see any objections
and questions to the author and the manuscript corresponds to the topic of SI: Design
and Synthesis of Macrocyclic Compounds.

Still the synthesis for the synthesis is not so interesting, so may be authors provides
some investigation of properties? Molecular recognition, selective extraction,
catalysis. From my point of view it will strongly enhance the manuscript.

Author Response

Point  1:  The paper is devoted to the synthesis of new macrocyclic compound with strong H-
bonding like protein-mimic. The manuscript describes synthesis and
characterization of four new macrocycles. In that part I do not see any objections
and questions to the author and the manuscript corresponds to the topic of SI: Design
and Synthesis of Macrocyclic Compounds.

Response 1:  Thank you very much for the support and your time spent on thoroughly examining our manuscript.

Point  2:  Still the synthesis for the synthesis is not so interesting, so may be authors provides
some investigation of properties? Molecular recognition, selective extraction,
catalysis. From my point of view it will strongly enhance the manuscript.

Response 2:  Thank you very much for the suggestions. Accordingly, we have added a preliminary molecular recognition study at the last part of the manuscript, which shows macrocycle 10 can recognize n-octyl-β-D-glucopyranoside (14) in chloroform. In the future, we will systematically screen diverse guest molecules as well as functional groups to achieve higher binding affinity and selective recognition, and the relevant results will be reported at an appropriate time.

Reviewer 4 Report

In manuscript “Recognition site modifiable macrocycle: synthesis, functional 2 group variation and structural inspection,” authors describe the synthesis of a biaryl macrocycle in gram scale which was functionalized using simple alkylation protocols. Moreover, NMR studies reveal important hydrogen signal shifts due to nuclear Overhauser effects which can be modulated by the before mentioned reactions.  These features provide insights about possible applications in molecular recognition. . Before publication, a minor revision is necessary:

-Authors should extend introduction part describing the synthesis and importance of previous biaryl macrocycles and providing appropriate references, for example: (a) J. Org. Chem. 2012, 77, 7, 3099 (b) Beilstein J. Org. Chem. 2009, 5, 71 (c) Org. Lett. 2000, 2, 22, 3477 (d) Molecules 2022, 27, 1012 (e) Angew. Che. Int. Ed. 2020, 59, 4835.

Author Response

Point  1:  In manuscript “Recognition site modifiable macrocycle: synthesis, functional 2 group variation and structural inspection,” authors describe the synthesis of a biaryl macrocycle in gram scale which was functionalized using simple alkylation protocols. Moreover, NMR studies reveal important hydrogen signal shifts due to nuclear Overhauser effects which can be modulated by the before mentioned reactions.  These features provide insights about possible applications in molecular recognition. . Before publication, a minor revision is necessary:

Response 1:  Thank you very much for the support and your time spent on thoroughly examining our manuscript.

Point  2:  Authors should extend introduction part describing the synthesis and importance of previous biaryl macrocycles and providing appropriate references, for example: (a) J. Org. Chem. 2012, 77, 7, 3099 (b) Beilstein J. Org. Chem. 2009, 5, 71 (c) Org. Lett. 2000, 2, 22, 3477 (d) Molecules 2022, 27, 1012 (e) Angew. Che. Int. Ed. 2020, 59, 4835.

Response 2:  Thank you very much for the kind suggestions. In the Introduction part, we added the importance of macrocyclic molecules in the pharmaceutical field, as well as examples of other biaryl-bridged macrocycles and their synthetic strategies. The corresponding references were also added in References section.